# AGG: Amortized Generative 3D Gaussians for Single Image to 3D

**Dejia Xu**  *dejia@utexas.edu*
*University of Texas at Austin*

**Ye Yuan**  *yey@nvidia.com*
*NVIDIA*

**Morteza Mardani**  *mmardani@nvidia.com*
*NVIDIA*

**Sifei Liu**  *sifeil@nvidia.com*
*NVIDIA*

**Jiaming Song**  *jiaming.tsong@gmail.com*
*NVIDIA*

**Zhangyang Wang**  *atlaswang@utexas.edu*
*University of Texas at Austin*

**Arash Vahdat**  *avahdat@nvidia.com*
*NVIDIA*

**Reviewed on OpenReview:** *https://openreview.net/forum?id=BOq3n5ewSP*

## Abstract

Given the growing need for automatic 3D content creation pipelines, various 3D representations have been studied to generate 3D objects from a single image. Due to its superior rendering efficiency, 3D Gaussian splatting-based models have recently excelled in both 3D reconstruction and generation. 3D Gaussian splatting approaches for image to 3D generation are often optimization-based, requiring many computationally expensive score-distillation steps. To overcome these challenges, we introduce an **A**mortized[1] **G**enerative 3D **G**aussian framework (**AGG**) that instantly produces 3D Gaussians from a single image, eliminating the need for per-instance optimization. Utilizing an intermediate hybrid representation, AGG decomposes the generation of 3D Gaussian locations and other appearance attributes for joint optimization. Moreover, we propose a cascaded pipeline that first generates a coarse representation of the 3D data and later upsamples it with a 3D Gaussian super-resolution module. Our method is evaluated against existing optimization-based 3D Gaussian frameworks and sampling-based pipelines utilizing other 3D representations, where AGG showcases competitive generation abilities both qualitatively and quantitatively while being several orders of magnitude faster. Project page: `https://ir1d.github.io/AGG/`

## 1 Introduction

The rapid development in virtual and augmented reality introduces increasing demand for automatic 3D content creation. Previous workflows often require tedious manual labor by human experts and need specific

---

[1] Our use of the term "amortized" originates from its original meaning: to gradually reduce or write off the initial cost of an asset over time. To avoid potential confusion, we would like to clarify that we are not directly employing the standard amortized optimization/inference framework.

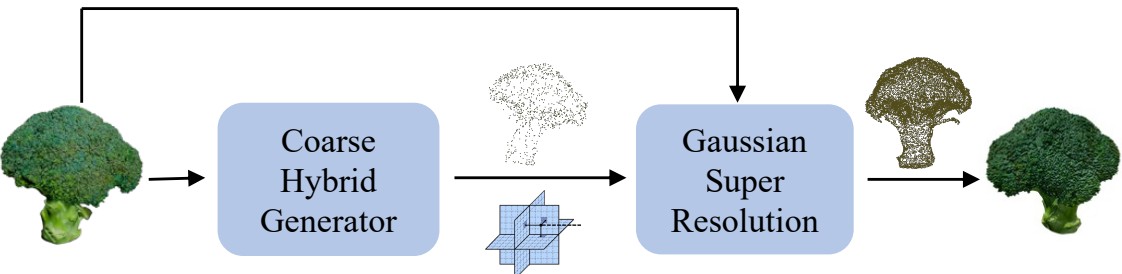

Figure 1: Overview of our AGG framework. We design a novel cascaded generation pipeline that produces 3D Gaussian-based objects without per-instance optimization. Our AGG framework involves a coarse generator that predicts a hybrid representation for 3D Gaussians at a low resolution and a super-resolution module that delivers dense 3D Gaussians in the fine stage.

software tools. A 3D asset generative model is essential to allow non-professional users to realize their ideas into actual 3D digital content.

Much effort has been put into image-to-3D generation as it allows users to control the generated content. Early methods study the generation of simple objects Yu et al. (2021) or novel view synthesis of limited viewpoints Li et al. (2021); Shih et al. (2020); Xu et al. (2022a). Later on, with the help of 2D text-to-image diffusion models, score distillation sampling Poole et al. (2022) has enabled 360-degree object generation for in-the-wild instances Xu et al. (2022b); Melas-Kyriazi et al. (2023); Tang et al. (2023b). More recently, the advances in collecting 3D datasets Deitke et al. (2023); Wu et al. (2023b); Yu et al. (2023) have supported the training of large-scale 3D-aware generative models Liu et al. (2023b;a); Gao et al. (2022) for better image-to-3D generations.

Various 3D representations have been studied as the media to train 3D generative models. Many prior works have explored generating explicit representations, such as point clouds Zeng et al. (2022), voxels Zhou et al. (2021); Schwarz et al. (2022), and occupancy grid Wu et al. (2023a). Implicit representations like NeRF Mildenhall et al. (2020) and distance functions Lionar et al. (2023) are popular for easy optimization. Although effective for content generation Yang et al. (2023); Erkoç et al. (2023); Or-El et al. (2022), slow rendering and optimization hinder the wide adoption of these representations. As a result, researchers have looked into more efficient volumetric representations Müller et al. (2022); Chan et al. (2022) to support the training of larger generative models Chen et al. (2023a); Karnewar et al. (2023) and the costly optimization in score distillation Xu et al. (2022b); Melas-Kyriazi et al. (2023). More recently, 3D Gaussian splatting Kerbl et al. (2023); Tang et al. (2023a); Chen et al. (2023b); Yi et al. (2023b) has attracted great attention due to its high-quality real-time rendering ability. This owes to the visibility-aware rendering algorithm that facilitates much less optimization time cost compared with NeRF-based variants Liu et al. (2023b); Poole et al. (2022). Such an efficient renderer enables supervising the 3D Gaussians through loss functions defined on the 2D renderings, which is not suitable for other representations.

Despite the exciting progress, how to properly generate 3D Gaussians Kerbl et al. (2023) remains a less studied topic. Existing works Tang et al. (2023a); Chen et al. (2023b); Yi et al. (2023b) focus on the per-instance optimization-based setting. Though 3D Gaussians can be optimized with adaptive density control to represent certain geometry, initialization remains critical for complex object structures. An amortized pipeline is highly desired to produce the 3D Gaussians in one shot. Such a network can learn a shared 3D understanding of images that generalizes to unseen objects of similar categories to the training set. This will further reduce the need for test-time optimization, trading off the computation cost of the inference stage with the training stage.

However, building such a feed-forward pipeline is challenging for two major reasons. First, 3D Gaussians rely on adaptive density control Kerbl et al. (2023) to represent complex geometry, but this leads to a dynamic number of 3D Gaussians, making it hard to predict them in an amortized training setting. Second, the 3D Gaussians require curated initialization Yi et al. (2023b); Chen et al. (2023b) to be updated properly

via supervision on the rendering. In the amortized setting, the 3D Gaussians are instead predicted by the neural network, which leads to the need to initialize the network well such that generated Gaussians can be supervised decently. Moreover, the optimization process often favors updating the appearance of 3D Gaussians instead of moving their positions directly to desired 3D locations.

To overcome these issues, we conduct a study on the amortized generation of 3D Gaussians through a feed-forward process. As shown in Fig. 1, we propose AGG, a cascaded generation framework that generates 3D Gaussians from a single image input. In the first stage, we employ a hybrid generator that produces 3D Gaussians at a coarse resolution. In this stage, we decompose the geometry and texture generation task into two distinct networks. A geometry transformer decodes image features extracted from a pre-trained image feature extractor and predicts the location of 3D Gaussians. Another texture transformer similarly generates a texture field that is later queried by the Gaussian locations to obtain other point attributes. Utilizing this hybrid representation as an intermediate optimization target stabilizes our training process when jointly optimizing the geometry and texture of the 3D Gaussians. In the second stage, we leverage point-voxel convolutional networks to extract local features effectively and super-resolve the coarse 3D Gaussians from the previous stage. RGB information is further injected into the super-resolution networks to refine the texture information.

Our contributions can be summarized as follows,

- We study single image-to-3D Gaussian in the amortized setting. Unlike existing works that operate on individual objects, we build a novel cascaded generation framework that instantly presents 3D Gaussians in one shot.

- Our AGG network first generates coarse Gaussian predictions through a hybrid representation that decomposes geometry and texture. With two separate transformers predicting the geometry and texture information, the 3D Gaussian attributes can be optimized jointly and stably.

- A UNet-based architecture with point-voxel layers is introduced for the second stage, which effectively super-resolves the 3D Gaussians.

- Compared with existing baselines including optimization-based 3D Gaussian pipelines and sampling-based frameworks that use other 3D representations, AGG demonstrates competitive performance both quantitatively and qualitatively while enabling zero-shot image-to-object generation, and being several orders of magnitude faster.

## 2 Related Works

### 2.1 Image-to-3D Generation

Numerous research studies have been conducted on generating 3D data from a single image. Early attempts simplify the challenges by either generating objects of simple geometry Yu et al. (2021) or focusing on synthesizing novel view images of limited viewpoints Li et al. (2021); Shih et al. (2020); Xu et al. (2022a). Later on, various combinations of 2D image encoder and 3D representations are adopted to build 3D generative models, such as on 3D voxels Girdhar et al. (2016); Choy et al. (2016); Yagubbayli et al. (2021), point clouds Yang et al. (2019); Fan et al. (2017); Zeng et al. (2022); Wu et al. (2023a); Lionar et al. (2023), meshes Gao et al. (2022); Kanazawa et al. (2018); Wang et al. (2018), and implicit functions Erkoç et al. (2023); Yang et al. (2021). More recently, with the help of score distillation sampling from a pre-trained text-to-image diffusion model, great efforts have been put into generating a 3D asset from a single image through optimization Xu et al. (2022b); Melas-Kyriazi et al. (2023); Seo et al. (2023); Tang et al. (2023b); Deng et al. (2023); Tang et al. (2023a). Among them, DreamGaussian utilizes 3D Gaussians as an efficient 3D representation that supports real-time high-resolution rendering via rasterization. In addition to these optimization-based methods, building feed-forward models for image-to-3D generation avoids the time-consuming optimization process, and a large generative model can learn unified representation for similar objects, leveraging prior knowledge from the 3D dataset better. The advances in large-scale 3D datasets Deitke et al. (2023); Wu et al. (2023b); Yu et al. (2023) largely contributed to the design of better

image-to-3D generative models Liu et al. (2023b;a); Gao et al. (2022); Hong et al. (2023); Li et al. (2023). OpenAI has trained a 3D point cloud generation model Point-E Nichol et al. (2022), based on millions of internal 3D models. They later published Shap-E Jun & Nichol (2023), which is trained on more 3D models and further supports generating textured meshes and neural radiance fields. Our work follows the direction of building feed-forward models for image-to-3D and makes the first attempt to construct an amortized model that predicts the 3D Gaussians instead of constructing them through optimization.

## 2.2 Implicit 3D Representations

Neural radiance field (NeRF) Mildenhall et al. (2020) implements a coordinate-based neural network to represent the 3D scenes and demonstrates outstanding novel view synthesis abilities. Many following works have attempted to improve NeRF in various aspects. For example, MipNeRF Barron et al. (2021a) and MipNeRF-360 Barron et al. (2021b) introduce advanced rendering techniques to avoid aliasing artifacts. SinNeRF Xu et al. (2022a), PixelNeRF Yu et al. (2021), and SparseNeRF Wang et al. (2023) extends the application of NeRFs to few-shot input views, making single image-to-3D generation through NeRF a more feasible direction. With the recent advances in score distillation sampling Poole et al. (2022), numerous approaches have looked into optimizing a neural radiance field through guidance from diffusion priors Xu et al. (2022b); Melas-Kyriazi et al. (2023); Seo et al. (2023); Tang et al. (2023b); Deng et al. (2023). While suitable for optimization, NeRFs are usually expressed implicitly through MLP parameters, which makes it challenging to generate them via network Erkoç et al. (2023). Consequently, many works Lorraine et al. (2023); Chan et al. (2022); Bhattarai et al. (2023) instead choose to generate hybrid representations, where NeRF MLP is adopted to decode features stored in explicit structures Chan et al. (2022); Chen et al. (2022); Müller et al. (2022); Fridovich-Keil et al. (2023); Yi et al. (2023a); Xu et al. (2022c); Barron et al. (2023). Among them, ATT3D Lorraine et al. (2023) builds an amortized framework for text-to-3D by generating Instant NGP Müller et al. (2022) via score distillation sampling. While focusing on generating explicit 3D Gaussians, our work draws inspiration from hybrid representations and utilizes hybrid structures as intermediate generation targets, which can be later decoded into 3D Gaussians.

## 2.3 Explicit 3D Representations

Explicit 3D representations have been widely studied for decades. Many works have attempted to construct 3D assets through 3D voxels Girdhar et al. (2016); Choy et al. (2016); Yagubbayli et al. (2021), point clouds Yang et al. (2019); Fan et al. (2017); Zeng et al. (2022); Wu et al. (2023a); Lionar et al. (2023) and meshes Gao et al. (2022); Kanazawa et al. (2018); Wang et al. (2018). Since pure implicit radiance fields are operationally slow, often needing millions of neural network queries to render large-scale scenes, great efforts have been put into integrating explicit representations with implicit radiance fields to combine their advantages. Many works looked into employing tensor factorization to achieve efficient explicit representations Chan et al. (2022); Chen et al. (2022); Fridovich-Keil et al. (2023); Yi et al. (2023a); Müller et al. (2022). Multi-scale hash grids Müller et al. (2022) and block decomposition Tancik et al. (2022) are introduced to extend to city-scale scenes. Another popular direction focuses on empowering explicit structures with additional implicit attributes. Point NeRF Xu et al. (2022c) utilizes neural 3D points to render a continuous radiance volume efficiently. Similarly, NU-MCC Lionar et al. (2023) uses latent point features as representation, specifically focusing on completing shapes. 3D Gaussian splatting Kerbl et al. (2023) introduces point-based $\alpha$-blending along with an efficient rasterizer and has attracted great attention due to their outstanding ability in reconstruction Kerbl et al. (2023) and generation Tang et al. (2023a); Chen et al. (2023b); Yi et al. (2023b) tasks. A concurrent work, Hu et al. (2024), addresses efficient point cloud rendering through a generalizable neural network converting point clouds to 3D Gaussians. However, existing 3D generation works present 3D Gaussians through optimization, while our work takes one step further and focuses on building an amortized framework that generates 3D Gaussians without per-instance optimization.

# 3 Amortized Generative 3D Gaussians

Unlike the existing methods that optimize Gaussians, initialized from structure-from-motion (SfM) or random blobs, our work pursues a more challenging objective: generating 3D Gaussians from a single image

using a neural network in one shot. Specifically, our cascaded approach begins with constructing a hybrid generator that maps input image features into a concise set of 3D Gaussian attributes. Subsequently, a UNet architecture with point-voxel layers is employed to super-resolve the 3D Gaussian representation, improving its fidelity. In the following subsections, we first provide the background on 3D Gaussian splatting, then we delve deep into details about our proposed components, and finally, we illustrate how we address the rising challenges with amortized generation.

### 3.1 Background: 3D Gaussian Splatting (3DGS)

3D Gaussian splatting is a recently popularized explicit 3D representation that utilizes anisotropic 3D Gaussians. Each 3D Gaussian $G$ is defined with a center position $\mu \in \mathbb{R}^3$, covariance matrix $\Sigma$, color information $c \in \mathbb{R}^3$ and opacity $\alpha \in \mathbb{R}^1$. The covariance matrix $\Sigma$ describes the configuration of an ellipsoid and is implemented via a scaling matrix $S \in \mathbb{R}^3$ and a rotation matrix $R \in \mathbb{R}^{3\times3}$.

$$\Sigma = RSS^T R^T \tag{1}$$

This factorization allows independent optimization of the Gaussians' attributes while maintaining the semi-definite property of the covariance matrix. In summary, each Gaussian centered at point (mean) $\mu$ is defined as follows:

$$G(x) = e^{-\frac{1}{2}x^T \Sigma^{-1} x}, \tag{2}$$

where $x$ refers to the distance between $\mu$ and the query point. $G(x)$ is multiplied by $\alpha$ in the blending process to construct the final accumulated color:

$$C = \sum_{i \in N} c_i \alpha_i G(x_i) \prod_{j=1}^{i-1}(1 - \alpha_j G(x_j)), \tag{3}$$

An efficient tile-based rasterizer enables fast forward and backward pass that facilitates real-time rendering. The optimization method of 3D Gaussian properties is interleaved with adaptive density control of Gaussians, namely, densifying and pruning operations, where Gaussians are added and occasionally removed. In our work, we follow the convention in 3D Gaussian splatting to establish the Gaussian attributes but identify specific canonicalization designed for single image-to-3D task. We first reduce the degree of the spherical harmonic coefficients and allow only diffuse colors for the 3D Gaussians. Additionally, we set canonical isotropic scale and rotation for the 3D Gaussians, as we find these attributes extremely unstable during the optimization process.

### 3.2 Coarse Hybrid Generator

In the coarse stage, our AGG framework utilizes a hybrid generator to produce a hybrid representation as an intermediate generation target. Initially, the input image is encoded using a vision transformer. Subsequently, a geometry and a texture generator are constructed to map learnable queries into location sequences and texture fields individually. This hybrid representation is then decoded into explicit 3D Gaussians, which enables efficient high-resolution rendering and facilitates supervision via multi-view images. The overall architecture of our proposed coarse hybrid generator is illustrated in Fig. 2. Our model is trained via rendering loss defined on multi-view images and is warmed up with Chamfer distance loss using 3D Gaussian pseudo labels.

**Encoding Input RGB Information** Single image-to-3D is a highly ill-posed problem since multiple 3D objects can align with the single-view projection. As a result, an effective image encoder is greatly needed to extract essential 3D information. Previous work Liu et al. (2023b;c) mainly incorporate CLIP image encoder Radford et al. (2021), benefiting from the foundational training of Stable diffusion Rombach et al. (2022). While effective for text-to-image generation, CLIP encoder is not well-designed for identifying feature correspondences, which is crucial for 3D vision tasks. As a result, we use the pre-trained DINOv2 transformer as our image encoder, which has demonstrated robust feature extraction capability through self-supervised pre-training. Unlike previous works Tumanyan et al. (2022); Xu et al. (2022a) that only use the aggregated global [CLS] token, we choose to incorporate patch-wise features as well.

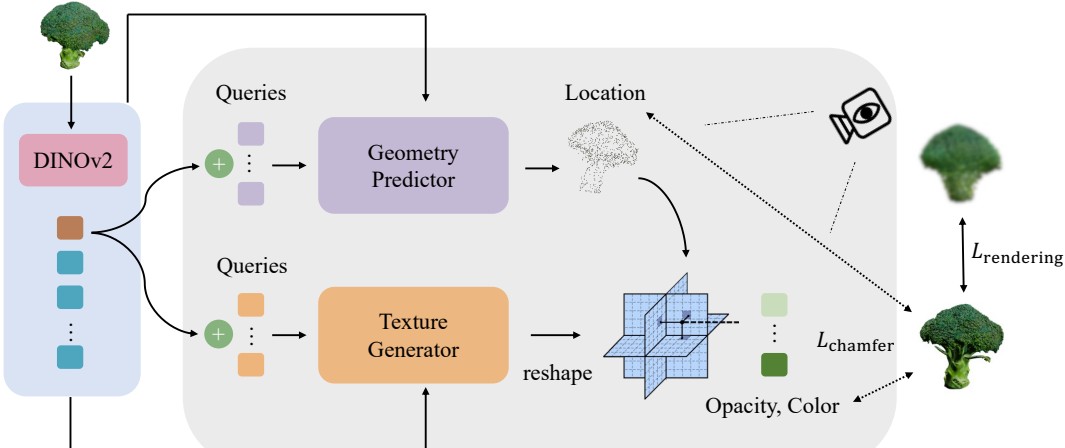

Figure 2: Architecture of our coarse hybrid generator. We first use a pre-trained DINOv2 image encoder to extract essential features and then adopt two transformers that individually map learnable query tokens to Gaussian locations and a texture field. The texture field accepts location queries from the geometry branch, and a decoding MLP further converts the interpolated plane features into Gaussian attributes (*e.g.*, opacity, color).

**Geometry Predictor** The location information of 3D Gaussian is expressed via their 3D means following 3DGS Kerbl et al. (2023), which is a three-dimensional vector for each point. Thus, we adopt a transformer-based network for predicting the location sequence. The transformer inputs are a set of learnable queries implemented with a group of learnable position embeddings. Each query will correspond to one 3D Gaussian that we generate. Before feeding into the transformer network, the positional embeddings are summed with the global token `[CLS]` extracted from the DINOv2 model. The sequence of queries is then modulated progressively by a series of transformer blocks, each containing a cross-attention block, a self-attention block, and a multi-layer perception block. These components are interleaved with LayerNorm and GeLU activations. The cross-attention blocks accept DINOv2 features as context information. The final layer of our geometry predictor involves an MLP decoding head, converting the hidden features generated by attention modules into a three-dimensional location vector.

**Texture Field Generator** Joint prediction of texture and geometry presents significant challenges. A primary challenge is the lack of direct ground truth supervision for texture in 3D space. Instead, texture information is inferred through rendering losses in 2D. When geometry and texture information are decoded from a shared network, their generation becomes inevitably intertwined. Consequently, updating the predicted location alters the supervision for rendered texture, leading to divergent optimization directions for texture prediction.

To resolve this, a distinct transformer is employed to generate a texture field. Our texture field is implemented using a triplane Chan et al. (2022), complemented by a shared decoding MLP head. The triplane accepts 3D location queries from the geometry branch and concatenates interpolated features for further processing. Utilization of this texture field facilitates decomposed optimization of geometry and texture information. More importantly, texture information is now stored in structured planes. Incorrect geometry predictions do not impact the texture branch's optimization process, which receives accurate supervision when geometry queries approximate ground truth locations.

**Supervision Through 2D Renderings** Thanks to the efficient rasterizer for 3D Gaussians, we can apply supervision on novel view renderings during the training. The 3D Gaussians we generate are rendered from randomly selected novel views, and image-space loss functions are calculated against ground truth renderings available from the dataset. Concretely, we render the scene into RGB images and corresponding foreground

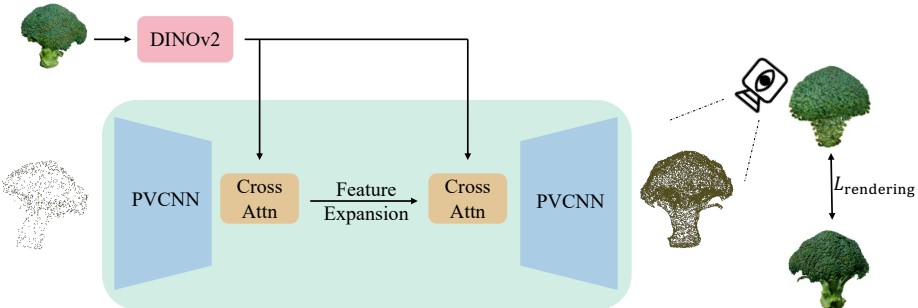

Figure 3: Illustration of the second-stage Gaussian super-resolution network. We first encode the original input image and the stage one prediction separately. Then, we unite them through cross-attention at the latent space. We perform super-resolution in the latent space and decode the features jointly.

alpha masks. We utilize LPIPS Zhang et al. (2018b) and $\mathcal{L}_1$ loss to minimize their differences,

$$\mathcal{L}_{\text{rendering}} = \mathcal{L}_{\text{rgba}} + \omega_1 \mathcal{L}_{\text{lpips}}, \tag{4}$$

where $\omega_1$ is a weighting factor.

### 3.3 Gaussian Super Resolution

Although our hybrid generator is effective in the coarse generation stage, generating high-resolution 3D Gaussians requires many learnable queries which are computationally expensive due to the quadratic cost of self-attention layers. As a result, we instead utilize a second-stage network as a super-resolution module to introduce dense 3D Gaussians into our generation. Since coarse geometry is obtained from the first stage, the super-resolution network can focus more on refining local details. For simplicity, we use a lightweight UNet architecture with the efficient point-voxel layers Liu et al. (2019), as shown in Fig. 3.

**Latent Space Super-Resolution**  As mentioned earlier, the 3D Gaussians require curated locations to update the other attributes properly. This leads to the key challenge for the super-resolution stage, which is to introduce more point locations for the super-resolved 3D Gaussians. Inspired by previous works in 2D image Shi et al. (2016) and point cloud super resolution Qian et al. (2021), we perform feature expanding as a surrogate of directly expanding the number of points. Through a point-voxel convolutional encoder, we first convert the stage one coarse 3D Gaussians into compact latent features that can be safely expanded through a feature expansion operation: rearranging a tensor of shape (B, N, C × r) to a tensor of shape (B, N × r, C). The expanded features are then decoded through a point-voxel convolutional decoder that predicts the Gaussian locations and other attributes.

**Incorporating RGB information**  While the first-stage network may capture the rough geometry of objects, the texture field may converge into blurry results due to the oscillating point locations and rough geometry. Consequently, utilizing the abundant texture information from the input image is crucial for the super-resolution network to generate plausible details. For this purpose, we introduce RGB features into the bottleneck of the UNet architecture. Specifically, we adopt cross-attention layers before and after the feature expansion operation. Image features are fed through cross-attention layers to modulate the latent point-voxel features.

### 3.4 Overcoming Challenges in Amortized Training

Unlike the vanilla 3D Gaussian splatting setting, where a set of 3D Gaussians are optimized specifically for each object, our amortized framework involves training a generator that jointly produces the 3D Gaussians for a large set of objects coming from diverse categories. As a result, many specifically designed operations

for manipulating 3D Gaussians are not available in our setting. Below, we illustrate how we overcome these challenges in detail.

### 3.4.1 Adaptive Density Control

The original 3D Gaussians involve special density control operations to move them toward their desired 3D locations. However, in the amortized training setting, it is non-straightforward to adaptively clone, split, or prune the Gaussians according to the gradients they receive. This is because these operations change the number of points, making it hard for an amortized framework to process.

To this end, we use a fixed number of 3D Gaussians for each object, avoiding the burden for the generator network to determine the number of points needed. Moreover, since we do not have access to clone and split operations, training the amortized generator leads to sharp 3D Gaussians with unpleasant colors that are trying to mimic fine-grained texture details. Once the predictions become sharp, they get stuck in the local minimum, and therefore, fewer 3D Gaussians are available to represent the overall object, leading to blurry or corrupted generations. To overcome the issue, we empirically set canonical isotropic scales and rotations for the 3D Gaussians on all objects to stabilize the training process.

### 3.4.2 Initialization

Proper initialization plays an important role in the original optimization-based 3D Gaussian splatting. Though Gaussian locations are jointly optimized with other attributes, it is observed that optimization updates often prefer reducing the opacity and scale instead of moving the Gaussians directly. By doing so, Gaussians are eliminated from the wrong location after reaching the pruning threshold. Instead, the adaptive density control can then clone or split Gaussians at proper locations according to their gradients. Albeit the simplicity of pruning and cloning operations for optimization algorithms, they are non-trivial to implement with amortized neural networks.

To overcome the issues, we warm up the generator with 3D Gaussian pseudo labels. We use a few 3D objects and perform multi-view reconstruction to obtain 3D Gaussians for each object. Since multiple sets of 3D Gaussians can all be plausible for reconstructing the same 3D ground truth object, the 3D Gaussians are only considered pseudo labels where we use their attributes to initialize our network layers properly.

Due to these Gaussians being stored in random orders as sets, we cannot use $\mathcal{L}_1$ reconstruction loss as in generative frameworks such as LION Zeng et al. (2022). Alternatively, we utilize the Chamfer distance loss to pre-train our network. Specifically, we first calculate the point matching correspondences according to the distance of Gaussians' 3D means and then minimize the $\mathcal{L}_1$ difference of each attribute, including location, opacity, and color. The formulation can be summarized as:

$$
\mathcal{L}_{\text{chamfer}}(f) = \frac{w_1}{|P_1|} \sum_{\substack{p_{1i} \in P_1, \\ p_{2j} = \arg\min_{x \in P_2}\left(\|p_{1i} - x\|_2^2\right)}} \left( \|f_{1i} - f_{2j}\| \right)
$$
$$
+ \frac{w_2}{|P_2|} \sum_{\substack{p_{2j} \in P_2, \\ p_{1i} = \arg\min_{x \in P_1}\left(\|x - p_{2j}\|_2^2\right)}} \left( \|f_{2j} - f_{1i}\| \right),
$$

where $f$ refers to 3D Gaussian attributes, $P_1$ is our generated set of 3D Gaussian, and $P_2$ is the 3D Gaussian pseudo label.

## 4 Experiments

### 4.1 Implementation Details

**Gaussian Configuration** We set the canonical isotropic scale of each 3D Gaussian to 0.03 and 0.01 when 4,096 and 16,384 3D Gaussians are produced for each object, respectively. The canonical rotation for all 3D

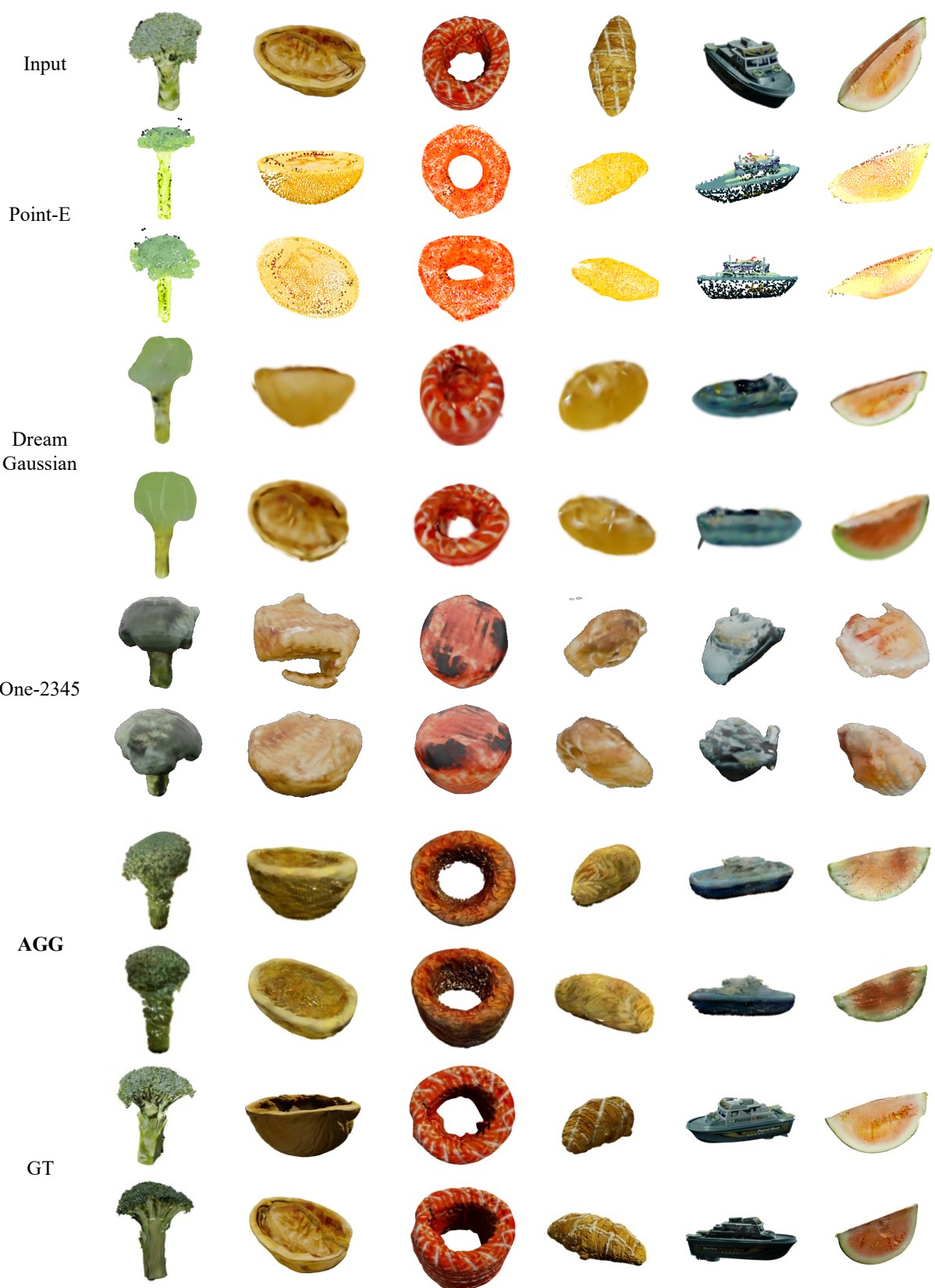

Figure 4: Novel view rendering comparisons against baseline methods. Our AGG model observes none of these testing images during training.

Gaussians is set to [1, 0, 0, 0] for the rasterizer. When preparing the pseudo labels for the warmup training, we use the ground truth point clouds from the OmniObject3D Wu et al. (2023b) dataset to initialize the Gaussian locations.

**Network Architecture**   We implement our transformer blocks following the well-established practice in DINOv2 Oquab et al. (2023) with GeLU activation and Layer Normalization layers. The image features are extracted using a pre-trained DINOv2-base model at $256 \times 256$ input resolution.

**Object Placement and Camera System**   We reconstruct objects in camera space and normalize the input view azimuth to zero. This means that the image and the 3D assets are aligned and consistent as both are rotated. This 3D augmentation during the training stage enforces rotation equivariance and prevents the network from merely overfitting the canonical placements of objects in the dataset.

## 4.2   Baseline Methods

We compare with two streams of work. One involves the sampling-based 3D generation that uses different 3D representations, such as Point-E and One-2345. Point-E Nichol et al. (2022) includes a large diffusion transformer that generates a point cloud. Their work performs the generation in the world coordinates, where objects are always canonically placed as the original orientation in the dataset. One-2345 Liu et al. (2023a) leverages Zero123 Liu et al. (2023b) and utilizes SparseNeuS Long et al. (2022) to fuse information from noisy multi-view generations efficiently. We also compare with DreamGaussian Tang et al. (2023a)'s first stage, where 3D Gaussians are optimized with SDS Poole et al. (2022) from Zero123 Liu et al. (2023b). Both One-2345 and DreamGaussian require the elevation of the input image at inference time. For One-2345, we use official scripts to estimate the elevation. We provide DreamGaussian with ground truth elevation extracted from the camera pose in the testing dataset.

## 4.3   Dataset

Our model is trained on OmniObject3D dataset Wu et al. (2023b), which contains high-quality scans of real-world objects. The number of 3D objects is much fewer than Objaverse Deitke et al. (2023). Alongside point clouds, the OmniObject3D dataset provides 2D renderings of objects obtained through Blender. This allows us to implement rendering-based loss functions that supervise the rendering quality in 2D space. We construct our training set using 2,370 objects from 73 classes in total. We train one model using all classes. The test set contains 146 objects, with two left-out objects per class.

## 4.4   Qualitative and Quantitative Comparisons

We provide visual comparisons and quantitative analysis between our methods and existing baselines in Fig. 4 and Tab. 1. As shown in the novel view synthesis results, our model learns reasonable geometry understanding and produces plausible generations of texture colors. Point-E Nichol et al. (2022) produces reasonable geometry but presents unrealistic colors, possibly because the model was trained on large-scale synthetic 3D data. Both One-2345 Liu et al. (2023a) and DreamGaussian Tang et al. (2023a) can generate corrupted shapes, which might be because the novel views synthesized from the Zero-123 diffusion model Liu et al. (2023b) are multi-view inconsistent.

We use CLIP distance defined on image embeddings to reflect high-level image similarity on multi-view renderings, as well as PSNR, SSIM, and LPIPS Zhang et al. (2018a) to measure the visual quality compared to the ground truth 3D object. For Point-E's outputs, we only use CLIP distance because they are produced in canonical orientation and not aligned with the ground truth 3D object. As shown in Tab. 1, our method obtains competitive numbers in terms of image quality. Additionally, our AGG network enjoys the fastest inference speed.

In comparison, both Point-E and One-2345 adopt an iterative diffusion process when producing their final results, while DreamGaussian leverages a costly optimization process through score distillation sampling.

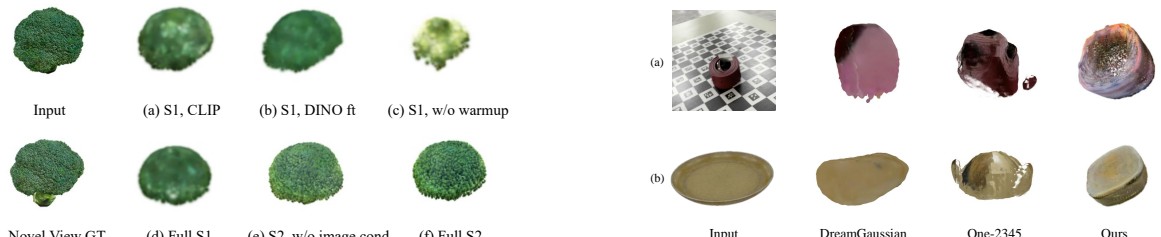

Figure 5: Visual comparison of ablation studies. Compared to variants (a,b,c,e), our full model in stage 1(d) and stage 2(f) brings better perceptual quality.

Figure 6: Comparison on an in-the-wild image (a) and Google Scanned Object (b).

Table 1: Comparisons against baseline methods regarding novel view image quality and inference speed.

| Method | Point-E | One-2345 | DreamGaussian | Ours |
|---|---|---|---|---|
| PSNR↑ | - | 14.73 | **18.87** | 18.36 |
| SSIM↑ | - | 0.7084 | **0.8147** | 0.8027 |
| LPIPS↓ | - | 0.3473 | 0.1906 | **0.1601** |
| CLIP Dist ↓ | 0.5139 | **0.3084** | 0.4293 | 0.3458 |
| Time (s)↓ | 78 | 45 | 60 | **0.19** |

Table 2: Ablation study of small-scale models that predict 2048 and 8192 Gaussians in stage 1 (S1) and stage 2 (S2). Models trained on 10 classes. Metrics were calculated on 20 held-out objects. Our full model in stage1(d) and stage2(f) brings better perceptual quality (LPIPS).

| Method | PSNR↑ | SSIM↑ | LPIPS↓ | Method | PSNR↑ | SSIM↑ | LPIPS↓ |
|---|---|---|---|---|---|---|---|
| (a) S1, CLIP | 15.82 | 0.7618 | 0.2440 | (d) Full S1 | 15.84 | 0.7619 | 0.2429 |
| (b) S1, DINO ft | 16.02 | 0.7615 | 0.2437 | (e) S2, w/o image cond | 15.91 | 0.7299 | 0.2322 |
| (c) S1, w/o warmup | 14.95 | **0.7631** | 0.2611 | (f) Full S2 | **16.03** | 0.7407 | **0.2217** |

## 4.5 Ablation Study

We conduct thorough experiments to validate the effectiveness of our proposed components using the available 3D ground truth from the dataset. We compare with multiple variants of the model. Our full model conditions on fixed DINO feature. We explored fixed CLIP features and fine-tuned DINO features in Fig. 5(a, b). In comparison, fixed DINO feature in Fig. 5(d) eases training, while other options degrade perceptual quality in the novel-view renderings. Then we study the effect of the warmup set in Fig. 5 (c). This set eases the optimization for geometry prediction. Without the warmup set, it is ineffective to rely on rendering loss for accurate geometry prediction. We observe a large drop in PSNR and LPIPS in Tab. 2(c) compared to (d). Finally, we remove the image conditioning in stage 2. As shown in Fig. 5 (e) and Tab. 2(e), image conditioning helps stage 2 model overcome unexpected color shifts. As shown in Tab. 2(f) and Fig. 5(f), the full model delivers the best generation results both qualitatively and quantitatively when rendered from novel viewpoints.

## 4.6 In-the-wild Evaluation

We further test our model on image from Google Scanned Objects Downs et al. (2022) and in-the-wild image, as shown in Fig. 6. The foreground object is segmented with the help of Segment Anything Model Kirillov et al. (2023). Compared to existing baselines, our model produces more reasonable 3D objects when provided

with out-of-distribution test cases. Existing works that rely on the view consistency of novel-view synthesis model fail to produce consistent results, therefore their merged 3D model contains unpleasant artifacts.

### 4.7 Limitation

Despite the encouraging visual results produced by AGG, the number of 3D Gaussians we generate is still limited to represent very complex geometry. In the future, we will further explore how to expand AGG to more challenging scenarios, such as when the input image contains multiple objects with occlusion.

## 5 Conclusion

In this work, we make the first attempt to develop an amortized pipeline capable of generating 3D Gaussians from a single image input. The proposed AGG framework utilizes a cascaded generation pipeline, comprising a coarse hybrid generator and a Gaussian super-resolution model. Experimental results demonstrate that our method achieves competitive performance with several orders of magnitude speedup in single-image-to-3D generation, compared to both optimization-based 3D Gaussian frameworks and sampling-based 3D generative models utilizing alternative 3D representations.

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

# A    Implementation Details

In this section, we provide more details about the implementation of our proposed AGG method.

## A.1    Training

The model is trained with Adam optimizer Kingma & Ba (2014). Rendering loss is enforced at $128 \times 128$ resolution. We first train the coarse hybrid generator for ten epochs. The learning rate is set to $1e - 4$ maximum, with a warmup stage for three epochs and followed by a cosine annealing learning rate scheduler. During the warmup epochs, we set the loss weight of $\mathcal{L}_{\text{chamfer}}$ to ten and $\mathcal{L}_{\text{rendering}}$ to one. Later, we gradually reduce the weight of $\mathcal{L}_{\text{chamfer}}$ and increase the weight of $\mathcal{L}_{\text{rendering}}$ until the weight of $\mathcal{L}_{\text{rendering}}$ is ten and that of $\mathcal{L}_{\text{chamfer}}$ is one. Then, we freeze the parameters for the coarse hybrid generator and only train the Gaussian super-resolution module. We use $\mathcal{L}_{\text{rendering}}$ and optimize for five epochs, with a learning rate set to $1e - 4$. Finally, we unfreeze all the parameters in both modules and train with rendering loss $\mathcal{L}_{\text{rendering}}$ for three epochs, with a learning rate set to $1e - 5$.

For $\mathcal{L}_{\text{rendering}}$, we set $\omega_1 = 2$ for all epochs. The parameters in DINOv2 Oquab et al. (2023) vision encoder are kept fixed in all epochs. The input image is $256 \times 256$ resolution, and the DINOv2 encoder produces image features of $1 \times 325 \times 768$. In each iteration, we render the generated 3D Gaussians into eight novel views where $\mathcal{L}_{\text{rendering}}$ are calculated against the ground truth correspondingly. The warmup set consists of ten objects per class, resulting in 730 objects in total.

## A.2    Inference

At inference time, given an in-the-wild image, we first extract image features using the DINOv2 encoder. Then, we send the features sequentially into the coarse hybrid generator and the Gaussian super-resolution network. Our model does not require access to the camera pose of the input image at inference time. This is because, during training, we ensure that the object is reconstructed in view space. In other words, the 3D asset and the input image are always aligned, and the supervision signals are rotated accordingly.

# B    Experiment Details

In this section, we provide more details about how the metrics are calculated, and results from the baselines are obtained.

## B.1    Metrics Calculation

Since the objects generated by the baseline methods are not guaranteed to be aligned with the ground truth 3D assets in the OmniObject3D Wu et al. (2023b) dataset, we cannot use pixel-wise reconstruction metrics to measure the generation quality. Moreover, the ill-posed nature of image-to-3D generation makes it harder to evaluate the 3D generation results quantitatively. Therefore, we follow the practice in previous works Xu et al. (2022b); Tang et al. (2023a) and leverage CLIP distance to measure the semantic quality of the generation results. Specifically, for each object, we first normalize the scale of the object, and then use a fixed set of cameras to obtain novel view renderings. Finally, we calculate the average spherical distance between the CLIP Radford et al. (2021) image embedding of the input image and the novel view images. The runtime of our method is measured on the test set using an A100 GPU. The runtime of baselines is obtained from DreamGaussian Tang et al. (2023a).

## B.2    Baseline Configurations

Point-E Nichol et al. (2022) consists of a transformer-based diffusion model that generates point clouds through sampling. We render the generated point cloud using Open3D Zhou et al. (2018) library and set the point size to 10. As for One-2345 Liu et al. (2023a), we render the generated mesh via the original Blender script provided by the authors. To render the results obtained by DreamGaussian Tang et al. (2023a), we use the author's script to first export the Gaussians to mesh and then render the mesh.

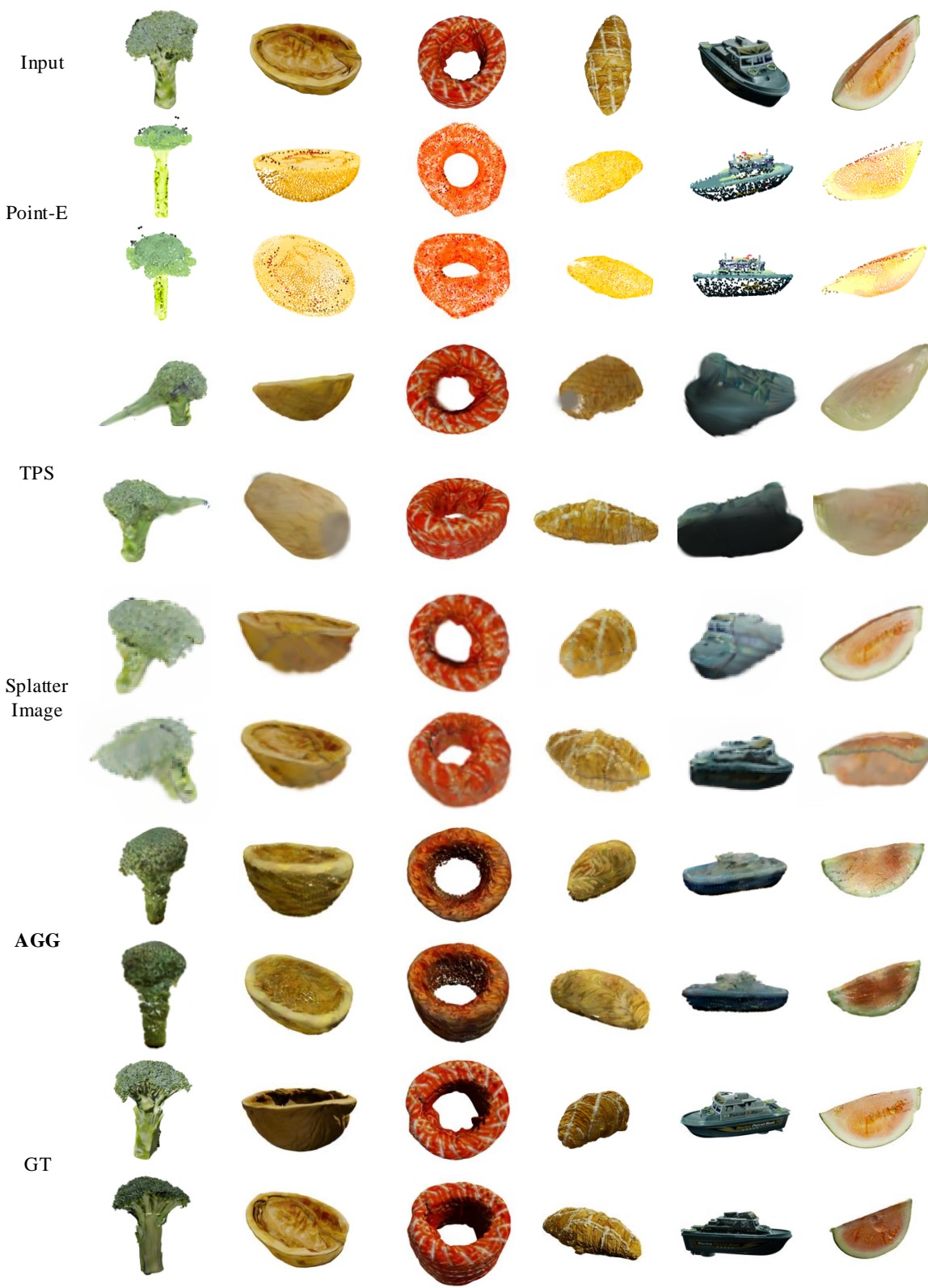

Figure A: Novel view rendering comparisons against concurrent works Splatter Image Szymanowicz et al. (2024) and TPS Zou et al. (2024). Our AGG model observes none of these testing images during training.

Table A: Additional comparisons with concurrent methods regarding novel view image quality.

| Method | Point-E | One-2345 | DreamGaussian | TPS | Splatter Image | Ours |
|---|---|---|---|---|---|---|
| PSNR↑ | - | 14.73 | 18.87 | 14.87 | 18.41 | 18.36 |
| SSIM↑ | - | 0.7084 | 0.8147 | 0.7548 | 0.8107 | 0.8027 |
| LPIPS↓ | - | 0.3473 | 0.1906 | 0.2602 | 0.1729 | 0.1601 |
| CLIP Dist ↓ | 0.5139 | 0.3084 | 0.4293 | 0.3804 | 0.3624 | 0.3458 |

## C  Video Results

**We provide more results and comparisons in the video. Please refer to the video for more details.**

## D  Additional Comparisons with Concurrent Works

As suggested by the Reviewer JHwb, we've added comparisons with concurrent works that appear in CVPR 2024, TPS Zou et al. (2024) and Splatter Image Szymanowicz et al. (2024). Please kindly note that these works are implemented independently.

To obtain the results from these concurrent methods, we use the authors' original implementations provided in their Github repos and render the generated 3D Gaussians directly. As shown in Tab. A, we quantitatively compare our proposed methods with other methods. Our work shows competitive performance in PSNR and SSIM, while outperforming most baselines in LPIPS and CLIP Distance that emphasize more on human perceptual quality.

We further provide side-by-side comparisons that qualitatively demonstrate our model's superiority. As shown in Fig. A, Splatter Image Szymanowicz et al. (2024) tends to produce results that are more blurry, and TPS Zou et al. (2024) suffers at generating realistic back-view results and side-view results. In comparison, our method outperforms these comparison methods in terms of geometry and appearance quality on challenging novel views.

## E  Broader Impacts

Existing 3D content creation pipelines usually require tedious work that an expert must do. On the positive side, the advent of image-to-3D generation frameworks lowers the barrier and makes 3D content creation accessible to a broader audience. Individuals and small businesses with limited resources can now produce high-quality 3D models without the need for extensive training or expensive software. This democratization can spur innovation, allowing more people to participate in industries such as gaming, virtual reality, education, and product design.

However, these easily generated 3D models could lead to a saturation of content, potentially devaluing the work of skilled professionals. Additionally, ethical considerations regarding the use of copyrighted or proprietary images to generate 3D models must be addressed. Ensuring that creators' rights are respected and that there is a clear framework for the ethical use of this technology is crucial.

