# OpenReview forum: "AGG: Amortized Generative 3D Gaussians for Single Image to 3D"
_TMLR — Accepted by TMLR_

### Review · Reviewer_JHwb · 2024-07-14

**Summary Of Contributions:**

* This paper introduces AGG (Amortized Generative 3D Gaussians), a feedforward method for generating 3D Gaussian representations from a single input image without per-instance optimization.
* The key idea is to use a cascaded architecture that first generates a coarse hybrid representation and then upsamples it to produce dense 3D Gaussians.
* The authors propose decomposing geometry and texture generation using separate transformer networks. A UNet model with point-voxel layers is then used to upsample the coarse 3D Gaussians.
* The method demonstrates competitive performance compared to optimization-based and sampling-based baselines while being orders of magnitude faster.

**Audience:**

Yes

**Broader Impact Concerns:**

It would be nice if the authors can add a Broader Impact Statement addressing potential ethical implications, including:
* Potential job displacement in industries relying on manual 3D modeling.
* Accessibility and democratization of 3D content creation, including both positive and negative implications.

**Claims And Evidence:**

No

**Requested Changes:**

I would appreciate if the authors can address Weaknesses #1 and #2 mentioned above and provide thorough results to address the same.

**Strengths And Weaknesses:**

## Strengths
1. **Novel approach**: The cascaded architecture for generating dense 3D Gaussians from a single input image is interesting, also making the 3D generation much faster.
    - The hybrid representation that decomposes geometry and texture generation is particularly clever, allowing for more stable joint optimization of 3D Gaussian attributes.
2. **Impressive speed improvement**: The method achieves orders of magnitude faster generation times compared to existing methods while maintaining competitive quality.
3. **Thorough ablations**: The paper includes comprehensive ablation studies clearly demonstrating contribution of different method components.

------------------

## Weaknesses
1. **Dependency on pre-trained encoder**: The reliance on a pre-trained DINOv2 image encoder may limit the method's ability to generalize to out-of-distribution images. It would be interesting to see how the method performs with different pre-trained encoders or with end-to-end training.
2. **Comparison with many single-image-to-3D works is missing**: I think the authors should quantitatively and qualitatively compare with the following single-image-to-3D methods for a complete comparison:
    - "Splatter Image: Ultra-Fast Single-View 3D Reconstruction", CVPR 2024
    - "Wonder3D: Single Image to 3D using Cross-Domain Diffusion", CVPR 2024
    - "Triplane Meets Gaussian Splatting: Fast and Generalizable Single-View 3D Reconstruction with Transformers", CVPR 2024
    - The following two methods should also be compared but are optional as one of them is only on arXiv and the second is from ECCV 2024:
        - "GRM: Large Gaussian Reconstruction Model for Efficient 3D Reconstruction and Generation", Xu et al.
        - "LGM: Large Multi-View Gaussian Model for High-Resolution 3D Content Creation", ECCV 2024
3. **Qualitative results don't look good**: This might be a subjective opinion, but for the examples shown in Figure 4 don't look impressive. Especially compared to Point-E (which I realize is a point-cloud method), AGG loses a lot of detail and results in blurry renderings.
    - I also think that the methods I listed above might give better or equally good looking outputs.

---

> ### Author Response · Authors · 2024-08-18
> **Author Response**
>
> We appreciate Reviewer JHwb for the positive assessment of our novelty, speed improvement, and thorough ablations. Below, we respond to your concerns in detail.
>
> **Q1. Dependency on the pre-trained encoder.**
>
> We have included an analysis of the image encoder in Fig 5. We explored using a fixed CLIP encoder (Fig. 5a) or a jointly trained DINO encoder (Fig. 5b). We observe that using a fixed DINO feature (Fig. 5d) eases training while other options degrade perceptual quality in the novel-view renderings.
>
> **Q2. Comparison with concurrent single-image-to-3D works.**
>
> In our updated revision, we have added comparisons with feed-forward Gaussian-based concurrent works ("Splatter Image" and "Triplane Meets Gaussian Splatting"), as suggested by the reviewer. Wonder3D, on contrary, first generates multi-view images, and then constructs 3D meshes using per-instance optimization. Note that these concurrent works are presented in CVPR 2024, which is after our initial submission. Our AGG framework is developed independently.
>
> As shown in Fig. A (page 18), our method generates less blurry results compared to these concurrent works at novel viewpoints. We observe in Tab. A (page 19) that, our method shows competitive performance in PSNR and SSIM, while outperforming most baselines in LPIPS and CLIP Distance that emphasize more on human perceptual quality.
>
> **Q3. Qualitative comparison with Point-E.**
>
> Please note that although Point-E shows impressive overall geometry quality, the appearance and geometry of its generated results largely differ from the input image. The main focus of image-to-3D generation, however, is to construct 3D objects that look like the input image. In comparison, our proposed AGG generates 3D objects with much better appearance quality, as shown in Fig. 4 and Fig. A.
>
> **Q4. Broader Impact Statement.**
>
> Thanks for your suggestions. We have included the broader impact statement in our revised draft (Section E), including discussions on potential job displacements and the accessibility and democratization of 3D content creation.

---

### Review · Reviewer_5dHr · 2024-07-20

**Summary Of Contributions:**

This paper presents a method to generate a collection of 3D Gaussians from a single image. Unlike existing works of 3D Gaussian splatting that requires per-scene optimization with ground-truth 2D views, the authors propose a feed-forward model that directly generates 3D Gaussian representations from a single image. Beyond existing work of 3D generation from single image, this work utilize the 3D Gaussian representation, allowing supervision in the rendering 2D space. The authors developed a coarse-to-fine geometry prediction pipeline and a separate texture triplane generation pipeline to achieve this goal. Experimental results demonstrate that the proposed method is capable of  generating more faithful and natural 3D structures than existing methods. In short, the main contribution of this paper is a proof of concept to generate 3D representation from 2D with a feed-forward 3D Gaussian generation neural network that shows better results.

**Audience:**

Yes

**Broader Impact Concerns:**

There is no ethical concerns on this paper.

**Claims And Evidence:**

Yes

**Requested Changes:**

The authors are encouraged to make the following changes,
1. Provide more details on how to combine the texture and geometry for rendering, as discussed in the first point of the weaknesses.
2. Add some discussion on the rendering visual quality of the generated scene by this method. Why the produced results are more blurry than the input images? How can future works improve?

**Strengths And Weaknesses:**

# Strengths
1. The idea of generating 3D Gaussians from single image with a feed-forward neural network is intuitive, and it is good that the authors show this idea can work, while they provide the technical configurations needed to make the system work.
2. The paper is overall well written and easy to understand, and the idea is clearly delivered. Despite that, some details are missing but I believe they can be easily added in a revision.
3. The proposed technique, which is shown to be capable of generating 3D objects from a single image, can potentially have a significant impact in both academia and industrial application. Nevertheless, it would be more convincing if the experiments can be conducted beyond simple objects as shown in the paper.

# Weaknesses
1. Some technical details are missing or are not made clear in the manuscript, Most importantly, the proposed method represents geometry and texture separately. However in 3D Gaussian rendering each Gaussian should be associated with a color value. It is not clear to me how the color is attached. I would guess there is a color retrieval process from the triplane to a 3D point but there is no clear description.
2. Despite quantitatively better numbers compared to existing method, the proposed method still renders pretty blurry methods, which is a bit far away from actually working in the real world. The method is mainly tested on simple and small objects, and there seems to be a lack of cross object category evaluation (namely all testing object types have been seen in the training). The results might not be encouraging enough for adoption in real applications.
3. The novelty in the technical aspect is not that significant. The feed-forward 3D Gaussian prediction and upsampling has already been explored in [1]. The other techniques like triplane representation and using pretrained models for feature extraction has also be widely explored.

[1] Hu, et al. "Low Latency Point Cloud Rendering with Learned Splatting." CVPRW, 2024.

---

> ### Author Response · Authors · 2024-08-18
> **Author Response**
>
> We appreciate Reviewer 5dHr’s positive assessment in our paper’s clear presentation and potential impact to the community. In regards to your questions, see our responses below:
>
> **Q1. How is the color attached? How to combine the texture and geometry for rendering?**
>
> Thanks for raising this question. As mentioned in Fig. 2, “The texture field accepts location queries from the geometry branch, and a decoding MLP further converts the interpolated plane features into Gaussian attributes.” Here the Gaussian attributes include the opacity and color information of the corresponding 3D Gaussians. We have clarified this in our revision (marked blue).
>
> The outputs of geometry predictor are the 3D locations of the 3D Gaussians. These locations are used to query features from the texture generator and converted to other Gaussian attributes. We obtain the texture and geometry features separately and send them to the rasterizer for rendering.
>
> **Q2. There is a lack of evaluation in real applications.**
>
> We have included in-the-wild evaluation on real unseen categories in Fig 6. Moreover, we train our model on OmniObject3D (Wu et al. 2023b), which contains high-quality scans of real-world objects. Though the object number is fewer than Objaverse, the geometry and texture quality of OmniObject3D are more challenging. Qualitative comparisons in Fig. 4 and Fig. A are also from real unseen testing images. These results show that our method outperforms existing and concurrent works from challenging novel viewpoints. Due to limited computational resources, we leave the exploration of scaling up the model for more object categories, such as Objaverse, to future work.
>
> **Q3. The novelty in the technical aspect is not that significant. Missing reference (Hu, et al. 2024).**
>
> Thanks for making us aware of the work by Hu, et al. We have cited and discussed this paper in our revised version (marked blue). However, Hu et al. are addressing the problem of efficient point cloud rendering while our focus is image-to-3D object generation. Note that this concurrent work is presented in CVPR 2024, which is after our initial submission. Our AGG framework is developed independently.
>
>
> **Q4. Why the produced results are more blurry than the input images? How can future works improve?**
>
> We appreciate this good question. As mentioned in Sec 4.7 Limitations, we use a limited number of 3D Gaussians to represent the 3D object due to the computation constraints. To further improve the texture and geometry quality, we will explore scaling up the network to work with more 3D Gaussians in future works. We may also explore self-supervised training techniques using real-world images, similar to Real3D[1].
>
> [1] Jiang H, Huang Q, Pavlakos G. Real3D: Scaling Up Large Reconstruction Models with Real-World Images[J]. arXiv preprint arXiv:2406.08479, 2024.

---

### Review · Reviewer_y2vy · 2024-08-06

**Summary Of Contributions:**

This paper works on reconstruct 3D gaussian represented objects from a single image. To this end, the authors proposed to use DINOv2 to extract image features at each patch, and then the patch feature token are incorporated into two transformers separately: first, is the geometry predictor that generate sparse gaussian locations point clouds, while the gaussian rotation and scale are manually fixed; second is the texture predictor that generates a tri-plane image features as the texture presentation; finally, the gaussian opacity and colors are decoded from interpolated tri-plane features via MLP at each guassian locations. To get higher density 3D gaussians, the authors adopt a point-voxel layer based U-Net to further upsample the sparse gaussians.

To demonstrate the effectiveness of the proposed method, the authors conduct quantitative comparison with existing works on OmniObject3D dataset, and achieve comparable results on various metrics with magnitude smaller time. The authors also conduct ablation studies to justify the importance of the DINOV2 features and the point cloud super-resolution.

**Audience:**

Yes

**Broader Impact Concerns:**

This paper works on reconstruct 3d models from a given single image, there is no broader impact concern as far as I can see.

**Claims And Evidence:**

Yes

**Requested Changes:**

Please refer to the weakness.

**Strengths And Weaknesses:**

Strengths:
- The paper introduces the problem well and sufficiently discusses the positioning of this work with respect to existing methods.

- The decoupled prediction is a well-designed approach, which can facilitate the single image reconstruction process by separately enforcing the photometric and geometric constraints.

- The authors discussed the insights for each design sufficiently. Specifically, in addition to the methods in Sections 3.2 and 3.3, the authors also discuss the challenges of reconstructing explicit 3D Gaussian distributions during training.

- The qualitative results are reasonably good.

Weakness:
- The method categorization is inprecise:
  - It is confusing to categorize the method as "amortized". Amortized optimization is not a widely acknowledged term and basically means "learning to optimize". However, the method proposed in this paper is a straightforward feed-forward inference without any optimization get involved. Specifically, the gaussians' scales and rotations are manually fixed, the spherical harmonics coefficients and opacity are predicted via MLPs from interpolated tri-plane features. To get optimization involved, there are several simple ways including but not limited to 1. the spherical harmonic coefficients can be optimized instead of predicted from MLPs, 2. after the feed-forward prediction, the gaussian can be refined using optimization, 3. and/or predict the output from features/context of gaussian optimization process such as the gradient of each quantity. It is better to either get optimization involved or find a better term than "amortized".
  - The method is not generative but deterministic. Specifically, the method does not adopt any generative model such as diffusion. Instead, it is better to replace "Generative 3D Gaussians" to "3D Gaussians Generation" since generation is a more general term that does not need to be generative model.
- The tri-plane image features are interpolated before sending to the MLPs. More elaboration is necessary about this interpolation process since it is critical. In Section 3.3, the authors mentioned that the appearance of gaussian can be blurry due to inaccurate point cloud locations. However, the blur can be also contributed by the interpolation, i.e. the MLP decoded tri-plane image features can produce tri-plane image can generate image with good quality, but after the interpolation it can be blurry. On the other hand, attention/correlation can be also considered other than vanilla interpolation, which can help the each gaussian retrieve the most relevant tri-plane image features and reduce the blur.
- Besides the rendering loss, the decoupled design can be better leveraged better by applying loss directly to the point cloud network and tri-plane image feature network. For example, we can constraint the point not to far from ground-truth surface and tri-plane image feature itself can produce high-quality tri-plane images.

---

> ### Author Response · Authors · 2024-08-18
> **Author Response**
>
> We appreciate Reviewer y2vy for the time and actionable suggestions. We are glad that the reviewer acknowledges our technical contributions and experimental results. Our responses to your concerns are listed as follows.
>
> **Q1. It is confusing to categorize the method as "amortized".**
>
> As mentioned in [1][2], “Amortized optimization methods use learning to predict the solutions to problems in these settings, exploiting the shared structure between similar problem instances.” Our AGG network is initially trained using amortized optimization. After the training is done, we perform cheap inference on new images.
>
> We agree with the reviewers in that our predictions can be refined using test time optimization. This doesn’t conflict with our AGG network being an amortized framework. In our paper, “Amortized” refers to the fact that image-to-3D conversion is done by a shared neural network. Instead of optimizing each 3D object separately at test time[3][4][5], this AGG network is trained using amortized optimization so we require no test time optimization.
>
> **Q2. The method is not generative but deterministic.**
>
> We agree that the method does not adopt any generative model such as diffusion. Our ‘generative’ refers to the task of 3D content creation compared with the task of 3D reconstruction. This term is commonly used in the community referring to content creation tasks[3][4][5]. We understand the confusion here and will be happy to clarify further in our revised version if the reviewer recommends it.
>
> **Q3. More elaboration on the triplane interpolation. How to overcome the blurry issue?**
>
> We appreciate this good question. Our geometry predictor outputs 3D locations of the 3D Gaussians. These 3D coordinates are used to query triplane features. The features are interpolated via `grid_sample` operations with `bilinear` mode. This interpolation is the same as the original triplane paper (Chan et al. (2022)) and is adopted in both training time and test time. In other words, each 3D location is already connected with the most relevant triplane features through interpolation.
>
> Our decomposed optimization of geometry and texture information avoids most blur cases since incorrect geometry predictions do not impact the texture branch’s optimization process. This ensures accurate supervision when geometry queries approximate ground truth locations. However, blurry results still exist mainly due to the capacity of the triplane features. To overcome this issue, we introduce a second-stage Gaussian Super Resolution module to introduce dense 3D Gaussians. We will try possible solutions (e.g. enlarging the triplane feature resolutions and introducing attention mechanisms as suggested by the reviewer) in future works.
>
> **Q4. Additional losses can be included to improve performance.**
>
> Thanks for mentioning these additional loss functions. In our proposed AGG framework, we also employ geometry supervision $L_{\text{chamfer}}$ to ensure the predicted 3D Gaussian locations are reasonable.
> We agree constraining points to be near surface and adding supervision to triplane features might help and we leave explorations to future works.
>
>
> [1] Amos B. Tutorial on amortized optimization[J]. Foundations and Trends® in Machine Learning, 2023, 16(5): 592-732.
>
> [2] Lorraine J, Xie K, Zeng X, et al. Att3d: Amortized text-to-3d object synthesis[C]//Proceedings of the IEEE/CVF International Conference on Computer Vision. 2023: 17946-17956.
>
> [3] Tang J, Ren J, Zhou H, et al. Dreamgaussian: Generative gaussian splatting for efficient 3d content creation[J]. arXiv preprint arXiv:2309.16653, 2023.
>
> [4] Ren J, Pan L, Tang J, et al. Dreamgaussian4d: Generative 4d gaussian splatting[J]. arXiv preprint arXiv:2312.17142, 2023.
>
> [5] Chen S, Zhou J, Jiang Z, et al. ScalingGaussian: Enhancing 3D Content Creation with Generative Gaussian Splatting[J]. arXiv preprint arXiv:2407.19035, 2024.

---

> > ### Comment · Reviewer_y2vy · 2024-09-22
> >
> > I appreciate the reply from the authors, however it does not address my concern clearly
> > - This paper works on single image based 3D generation which does not have an optimization formulation itself. Although there is 3D Gaussian Splatting adopted in the paper, the 3D gaussian splatting is conducted on the ground-truth multi-view images. The generated 3D Gaussians are only the representation of the model prediction, while the inference itself is not solving a 3D Gaussian Splatting optimization process. The ground-truth multi-view images are invalid during inference and can not be considered as the "context" or input of the optimization problem (if there is any) the inference trying to solve.
> > To make it clear, the authors should first define what is the optimization problem here, which is not 3D Gaussian Splatting on multi-view images, and how this optimization is amortized.
> > - In general, I highly recommend the authors to replace the misleading term "amortized" due to the problem inconsistency between 3D Gaussian Splatting and single image 3D reconstruction.

---

> > > ### Author Response · Authors · 2024-09-22
> > >
> > > Dear Reviewer y2vy,
> > >
> > > Thank you for your valuable feedback and thoughtful comments. We would like to respectfully address some concerns you raised.
> > >
> > > You stated, “This paper works on single image-based 3D generation, which does not have an optimization formulation itself.” We believe this assertion is inaccurate. Previous works that utilize 3D Gaussians for 3D generation [1][2][3] are optimization-based and involve a computationally intensive score distillation process to optimize a 3D object during inference. For these tasks, even though ground truth multi-view images are unavailable, 3D Gaussians are optimized at inference time using learnable priors.
> > >
> > > In contrast, our approach eliminates the need for this time-consuming, per-instance optimization by introducing a feedforward framework that generates 3D Gaussians directly from a single image at test time.
> > >
> > > Additionally, the term “amortized” in our work is consistent with its usage in the ICCV paper "ATT3D: Amortized Text-to-3D Object Synthesis" [4]. That paper addresses the per-instance optimization issue in a text-to-3D generation context using a NeRF-based framework. Our work similarly tackles this issue but focuses on image-to-3D generation with 3D Gaussians.
> > >
> > > In summary, our use of the terms “amortized” and “optimization” in the context of 3D generation is consistent with established terminology in the research community. We hope this clarification can help address your concerns. Please do not hesitate to contact us if there are other clarifications we can provide. Thanks!
> > >
> > > Sincerely,
> > >
> > >
> > > The Authors
> > >
> > > [1] Tang J, Ren J, Zhou H, et al. Dreamgaussian: Generative gaussian splatting for efficient 3d content creation[J]. arXiv preprint arXiv:2309.16653, 2023.
> > >
> > > [2] Yi T, Fang J, Wang J, et al. Gaussiandreamer: Fast generation from text to 3d gaussians by
> > > bridging 2d and 3d diffusion models[C]//Proceedings of the IEEE/CVF Conference on Computer Vision and Pattern Recognition. 2024: 6796-6807.
> > >
> > > [3] Chen Z, Wang F, Wang Y, et al. Text-to-3d using gaussian splatting[C]//Proceedings of the IEEE/CVF Conference on Computer Vision and Pattern Recognition. 2024: 21401-21412.
> > >
> > > [4] Lorraine J, Xie K, Zeng X, et al. Att3d: Amortized text-to-3d object synthesis[C]//Proceedings of the IEEE/CVF International Conference on Computer Vision. 2023: 17946-17956.

---

> ### Comment · Reviewer_y2vy · 2024-09-22
>
> - Single image as input to the pipeline does not mean that the single image reconstruction can be formulated as an optimization problem. Both Nerf and 3D Gaussians need to be optimized on multi-views, which need to be sampled in the space.
> - This paper has a fundamental difference from [4], where the training involves an optimization on the (rendered) multi-view output from the Nerf with predicted parameters. The nerf parameter optimization is amortized  as a feedforward inference. While in this paper, the 3D Gaussian splatting is conducted on ground-truth, it is only a pre-processing that convert the data to the desired 3D Gaussian representation from ground-truth multi-view images.
> - I would suggest the author refer to "Tutorial on amortized optimization" directly, follow the definition, and give the problem formulation strictly based on the definition to avoid any confusion.

---

> > ### Author Response · Authors · 2024-09-22
> >
> > Dear Reviewer y2vy,
> >
> > Thank you for your prompt response. We understand your concerns and would like to clarify the following points:
> >
> > Our work involves optimizing the multi-view renderings of the predicted 3D Gaussians. Importantly, our model does not overfit the pre-processed 3D Gaussians (3DGS). While we use pseudo-labels, they represent only a small portion of the training set and serve primarily to initialize and regularize the model's predictions via Chamfer distance. The bulk of the training relies on supervision from 2D renderings of the predicted 3D Gaussians, as shown in Eq. 1.
> >
> > Although optimizing NeRF and 3D Gaussians on multi-view data is ideal, as demonstrated in prior works [1][2][3][4], NeRF/3D Gaussians from a single input image are typically obtained through per-instance optimization. In contrast, our approach amortizes the optimization of 3DGS parameters.
> >
> > Please feel free to let us know if there are other clarifications we can provide.
> >
> > Thanks,
> >
> > Authors
> >
> >
> > [1] Tang J, Ren J, Zhou H, et al. Dreamgaussian: Generative gaussian splatting for efficient 3d content creation[J]. arXiv preprint arXiv:2309.16653, 2023.
> >
> > [2] Yi T, Fang J, Wang J, et al. Gaussiandreamer: Fast generation from text to 3d gaussians by bridging 2d and 3d diffusion models[C]//Proceedings of the IEEE/CVF Conference on Computer Vision and Pattern Recognition. 2024: 6796-6807.
> >
> > [3] Chen Z, Wang F, Wang Y, et al. Text-to-3d using gaussian splatting[C]//Proceedings of the IEEE/CVF Conference on Computer Vision and Pattern Recognition. 2024: 21401-21412.
> >
> > [4] Lorraine J, Xie K, Zeng X, et al. Att3d: Amortized text-to-3d object synthesis[C]//Proceedings of the IEEE/CVF International Conference on Computer Vision. 2023: 17946-17956.

---

> > > ### Comment · Reviewer_y2vy · 2024-09-22
> > >
> > > Dear Authors:
> > >   Thank you for the swift response.
> > >   Unfortunately, it does not answers my question directly.
> > >   - "Our work involves optimizing the multi-view...": This statement is consistent with my concern about there is is no optimization amortized. If 3D Gaussian Splatting is not the optimization to be amortized, what is it then?
> > >   - "...our approach amortizes the optimization of 3DGS parameters...": This statement is contradict to the above one. What does the optimization refer to here? It should not be the 3D Gaussian Splatting process itself.
> > >   - "NeRF/3D Gaussians from a single input image are typically obtained through per-instance optimization." Again, generating 3D representation from single image does not mean that there is an optimization defined on single image. Before discussing whether the optimization is amortized, the author should first answer the question what is this optimization, which is obviously not 3D Gaussian Splatting process.
> > >   - To save the effort for further discussion, please strictly follow the formulation in "Tutorial on amortized optimization" and give the definition of the problem. It is more important to make the problem clear and precise rather than obsessing with specific terms.

---

> > > > ### Author Response · Authors · 2024-09-22
> > > >
> > > > Dear Reviewer y2vy,
> > > >
> > > > Thank you for your prompt reply. We recognize that some of our previous statements may have been unclear, and we would like to provide further clarification.
> > > >
> > > > You noted in the statement: _"Generating a 3D representation from a single image does not mean that there is an optimization defined on the single image… the author should first answer the question of what this optimization is, which is obviously not the 3D Gaussian Splatting process."_
> > > >
> > > > Your concern seems to stem from a potential distinction between 3D Gaussian Splatting (3DGS) constructed from multi-view image input, which you might interpret as involving optimization, and 3DGS from single-view image input[1][2][3], which you might perceive as not involving optimization.
> > > >
> > > > To clarify, the "optimization" we refer to here involves optimizing the 3D Gaussian parameters, which can be supervised by rendering loss, score distillation loss, or other similar losses. This optimization occurs **regardless of the number of input images**. In previous works [1][2][3], this per-instance optimization happens **during inference**. However, in our work, our framework **predicts** the 3D Gaussian parameters, supervised by rendering loss **at training time**, allowing us to eliminate the need for per-instance optimization at test time. As a result, our model can directly predict the 3DGS parameters without requiring additional optimization during inference.
> > > >
> > > > We hope this explanation addresses your concern. Please do not hesitate to reach out if further clarification is needed.
> > > >
> > > > Thanks,
> > > >
> > > > Authors.

---

> > > > > ### Comment · Reviewer_y2vy · 2024-09-22
> > > > >
> > > > > Dear Authors:
> > > > >   - Per-instance optimization is __replaced__ with a feedforward inference is different from per-instnace optimization is amortized. Using [4] as an example, during training SDS loss is defined on the rendered views from the Nerf and the diffusion model, which is how original DreamFusion optimize the Nerf parameters. While the rendering loss in this paper is defined on the rendered 3D Gaussians and __ground-truth__, which is not how 3D Gaussians are optimized in the context of single image reconstruction. We can not consider the rendering loss is doing the job of optimization (it is only a supervised training), unless the rendering loss is defined on some generated views instead of __ground-truth__. But I need to highlight that I am not suggesting to do so, it is only to clarify why this paper is not amortized optimization.
> > > > >  -  Again, to save the effort for further discussion, please strictly follow the formulation in "Tutorial on amortized optimization" and give the definition of the problem. It is more important to make the problem clear and precise rather than obsessing with specific terms.

---

> ### Author Response · Authors · 2024-09-24
>
> Dear Reviewer y2vy,
>
> Thank you for clarifying your perspective on this. As an example, DreamGaussian[1] performs per-instance optimization by distilling the Zero-123 diffusion model into 3D Gaussians. Their diffusion model is pre-trained on multi-view RGB images, and the per-instance optimization aims to minimize the KL divergence between the images rendered by 3D Gaussian representations and the multi-view ground truth.
>
> Here, our approach offers an alternative by replacing the per-instance optimization with a feed-forward network. The use of the amortization term in our method was originally inspired by the amortized inference literature, where optimization-based inference in variational inference is replaced with feed-forward networks, e.g. in VAEs [5]. We are happy to clarify this point in the revised version.
>
> [5] Margossian C C, Blei D M. Amortized Variational Inference: When and Why?[J]. arXiv preprint arXiv:2307.11018, 2023.

---

> ### Comment · Reviewer_y2vy · 2024-09-24
>
> - "As an example, DreamGaussian[1] performs..." This statement from authors is a counterexample that explain the issue in this paper: there is no GaussianSplatting process defined directly on ground-truth in DreamGaussian, and it is the case of "loss is defined on some generated views instead of ground-truth" as in my previous reply.
>
> - For both amortized optimization and amortized inference, the basic requirement for both of them is that the input in original version and the amortized version should be consistent. In addition, I do not see there is any relevance to amortized inference here since this paper is not a generative model.
>
> - For the last time, please give the problem formulation here based on the definition, therefore why "amortized" is misleading will be clear.

---

> > ### Comment · Action_Editor_cKWn · 2024-09-24
> >
> > Dear authors and reviewer y2vy,
> >
> > Thank you for the discussion, from which it seems the major concern is about the correct usage of the term *amortized* and the corresponding definition.
> >
> > Please make sure the definition is correct and address any potential misleading terms, before this paper can be considered for publication.
> >
> > Reviewer **JHwb** and **5dHr**, please feel free to share your thoughts as well.
> >
> >
> > Thanks,
> >
> > AE

---

> > ### Author Response · Authors · 2024-09-24
> >
> > Dear AE and Reviewer y2vy,
> >
> > Thank you to the reviewer for sharing their perspective. To avoid potential confusion, we have clarified in our revised version that we are not directly employing the standard amortized optimization/inference framework. A footnote, marked in blue, has been added to the abstract. Additionally, we would like to note that our use of the term "amortized" aligns with its original meaning: to gradually reduce or write off the initial cost of an asset over time. In our framework, the cost of test-time optimization is shifted to the training stage.

---

### Comment · Action_Editor_cKWn · 2024-08-12
**The discussion phase has begun**

Dear Authors,

Now that the review comments from all three expert reviewers have been released, please be reminded that you may start the discussion with the reviewers by posting rebuttals and updating your paper in response to the reviews.


Thanks,

AE

---

> ### Author Response · Authors · 2024-08-18
> **Author Response**
>
> Dear AE,
>
> Thank you for your continued support in overseeing the review process. We sincerely appreciate the insightful reviews from the reviewers, which have been instrumental in improving our submission.
>
> Based on their insightful comments, we have incorporated the suggested experiments and elaborated on our methods. New materials are marked blue in our revision. We eagerly await the reviewers’ feedback on our response. If there are any ambiguities or further questions, we are more than willing to provide clarity or delve deeper into any topic.
>
> Your continued guidance is greatly appreciated.
>
> Warm regards,
>
> Authors of Paper 2867.

---

> > ### Comment · Action_Editor_cKWn · 2024-08-19
> >
> > Dear Reviewers,
> >
> > Thank you again for your contributions to the reviewing of this submission.
> >
> > Now the authors have submitted their responses to your comments and a revised version of their manuscript. Please take a look at them see if they addressed your concerns and if you have any further concerns/questions.
> >
> > Please start a discussion with the authors wherever needed. The authors are also suggested to participate in the discussion actively to address further concerns if any.
> >
> > Best,
> >
> > AE

---

### Decision · Action_Editor_cKWn · 2024-09-29

**Recommendation:** Accept with minor revision

**Comment:**

In this paper, the authors presented a generative 3D Gaussian framework (named AGG) to address the problem of 3D from a single image. The 3D Gaussian location generation and other appearance attributes are decomposed, and a pipeline of coarse representation generation followed by upsampling is proposed. Experimental results against other 3D Gaussian methods show the effectiveness of the proposed method. The paper is generally well-written.

Three expert reviewers were invited to review the paper, and both strengths and weaknesses were raised. The paper received two Accept and one Reject recommendations. The major concerns include the use of the term "amortized", the decoupled design, unclear technical details, novelty, and experiments (e.g. missing comparison to related work, and poor qualitative performance). After the authors' response and revision, most concerns were addressed, with a remaining one about the potential misleading usage of the term "amortized". With back-and-forth discussions and revision, this became more clear and the authors promised to address the misleading issue by further clarification and clear definition. On the other hand, the reviewers and AE found the techniques and results presented in this paper might be of interest to a group of audiences (e.g. the 3D generation community) in TMLR. As a result, the AE would be happy to recommend Accept (with minor revision).

The authors are suggested to clearly clarify the potential misleading usage of the term mentioned above, throughout the whole paper. If possible, please also try to avoid using the term "amortized" too much and not highlighting it, as it is not considered to be a critical point of the proposed method but potentially lead to misleading.

**Audience:**

Yes, at least some individuals in TMLR's audience will be interested in knowing the findings of this paper.

**Claims And Evidence:**

The claims made in the submission are mostly supported by accurate, convincing and clear evidence, except the potential misleading usage of the term "amortized".